# Single-Step Synthesis of Graphitic Carbon Nitride Nanomaterials by Directly Calcining the Mixture of Urea and Thiourea: Application for Rhodamine B (RhB) Dye Degradation

**DOI:** 10.3390/nano13040762

**Published:** 2023-02-17

**Authors:** Agidew Sewnet, Esayas Alemayehu, Mulualem Abebe, Dhakshnamoorthy Mani, Sabu Thomas, Nandakumar Kalarikkal, Bernd Lennartz

**Affiliations:** 1Faculty of Materials Science and Engineering, Jimma University, Jimma P.O. Box 378, Ethiopia; 2Department of Physics, College of Natural and Computational Science, Bonga University, Bonga P.O. Box 334, Ethiopia; 3Faculty of Civil and Environmental Engineering, Jimma University, Jimma P.O. Box 378, Ethiopia; 4School of Chemical Sciences, Mahatma Gandhi University, Kottayam 686560, India; 5School of Pure and Applied Physics, Mahatma Gandhi University, Kottayam 686560, India; 6Faculty of Agricultural and Environmental Sciences, University of Rostock, Justus-Von-Liebig-Weg 6, 18059 Rostock, Germany

**Keywords:** g-C_3_N_4_, calcination temperatures, rhodamine B, photocatalytic dye degradation

## Abstract

Recently, polymeric graphitic carbon nitride (g-C_3_N_4_) has been explored as a potential catalytic material for the removal of organic pollutants in wastewater. In this work, graphitic carbon nitride (g-C_3_N_4_) photocatalysts were synthesized using mixtures of low-cost, environment-friendly urea and thiourea as precursors by varying calcination temperatures ranging from 500 to 650 °C for 3 h in an air medium. Different analytical methods were used to characterize prepared g-C_3_N_4_ samples. The effects of different calcination temperatures on the structural, morphological, optical, and physiochemical properties of g-C_3_N_4_ photocatalysts were investigated. The results showed that rhodamine B (RhB) dye removal efficiency of g-C_3_N_4_ prepared at a calcination temperature of 600 °C exhibited 94.83% within 180 min visible LED light irradiation. Photocatalytic activity of g-C_3_N_4_ was enhanced by calcination at higher temperatures, possibly by increasing crystallinity that ameliorated the separation of photoinduced charge carriers. Thus, controlling the type of precursors and calcination temperatures has a great impact on the photocatalytic performance of g-C_3_N_4_ towards the photodegradation of RhB dye. This investigation provides useful information about the synthesis of novel polymeric g-C_3_N_4_ photocatalysts using a mixture of two different environmentally benign precursors at high calcination temperatures for the photodegradation of organic pollutants.

## 1. Introduction

Nowadays, growing industrial activities have resulted in the generation of huge amounts of hazardous pollutants that are released into environmental water bodies [1,2]. The major causes of water pollution are the release of untreated industrial effluents and uncontrolled anthropogenic activities [3,4,5]. Meanwhile, various hazardous pollutants including antibiotics, phenols, pesticides, pharmaceuticals, and dyes from textile industries are causes of major water pollution [6]. Among them, dyes from textile industries generate a large amount of industrial wastewater due to the huge water demand and frequent discharge into the environment [6,7]. In addition, approximately 10–20% of total dyes from textile industries in the world are dissipated as effluent into the environment throughout the production process [2,8,9]. Therefore, the random and untreated discharge of these hazardous pollutants causes significant water pollution and has a direct impact on human health [2]. As a result, water pollution remediation is one of the imperative issues that the scientific community has given significant attention to with the primary goal of safeguarding and conserving natural water resources [10]. To date, wastewater has been treated using a variety of physical, chemical, and biological techniques, but the scientific community has yet to find a method that is renewable and sustainable [11,12]. Furthermore, these conventional methods have drawbacks, such as requiring a large capital investment, low effectiveness, expensive maintenance costs, and requiring large operating areas, and also cannot fully eliminate hazardous organic compounds beyond changing their phase. Therefore, it is essential to design efficient, sustainable, and renewable wastewater treatment technologies for effective removal of hazardous organic pollutants [2,11,12,13]. Among the several known advanced oxidation technologies, semiconductor-based heterogeneous photocatalysis is the most efficient advanced oxidation process to degrade pollutants at ambient pressure and temperature without generating harmful intermediates [11]. In the photocatalytic process, light is absorbed by the semiconductor material and initiates the photocatalytic reaction. The rate of photocatalytic reaction depends on the ability of the photocatalyst to generate electron-hole pairs, the specific surface area, the crystallinity, the spectral response, and the light absorption capacity [14]. An efficient photocatalyst should meet a variety of requirements, including excellent optical response, tunable band gap energy, superior photochemical stability, and high affinity for light [15]. Therefore, photocatalytic degradation of organic pollutants has become one of the most promisingly sustainable, cost-effective, green, and clean technology [13,16,17]. In the past few decades, metal oxide-based photocatalysts such as TiO_2_, ZnO, and SnO_2_ have been exploited in the remediation of environmental pollution. Nevertheless, the substantial electron-hole pair recombination in these photocatalysts significantly decreases their photocatalytic efficiency. Furthermore, ZnO, TiO_2_, and SnO_2_ can absorb solar energy exclusively in the UV region because of their wide energy bandgap, which accounts for just 4% of the total solar energy irradiated. Nowadays, because of its fascinating physicochemical properties, g-C_3_N_4_ is a superior alternative for visible light-assisted photocatalytic wastewater treatment than other photocatalysts such as TiO_2_, ZnO, SnO_2,_ and so on [18].

The majority of photocatalysis research has used xenon lamps [19,20], halogen lamps [21], UV lamps [22], and mercury vapor lamps [23] as the light source, which has some advantages but is dangerous, has a short lifetime, overheats, is expensive and difficult to handle [24]. Recently, in the field of photocatalysis technology, less toxic, longer life, less expensive, and environmentally friendly light-emitting diodes have been employed as visible light sources for organic pollutant degradation [24,25].

Since its discovery by Wang’s group in 2009, g-C_3_N_4_ has piqued the interest of researchers as a visible-light-driven semiconductor photocatalyst [26]. Polymeric g-C_3_N_4_ is an extensively explored n-type semiconductor for the removal of organic pollutants due to its inexpensiveness, non-toxicity, eco-friendliness, simple synthesis, mid-energy bandgap, and robust physicochemical stability [2,27]. In general, g-C_3_N_4_ can be synthesized by thermal calcination of various nitrogen-rich precursors, such as cyanamide [28], melamine [29], dicyandiamide [30], thiourea [31], and urea [20,32].

Nevertheless, pristine g-C_3_N_4_ suffers from a low photocatalytic performance due to the rapid recombination of photogenerated charge carriers, limited range absorption of visible light, and low specific surface area [33]. Therefore, researchers have made numerous attempts to overcome the limitations of pristine g-C_3_N_4_, such as elemental doping [27,34,35,36], controlling morphology [37], and constructing g-C_3_N_4_-based heterojunctions [38,39,40]. Moreover, recently various studies have been conducted on g-C_3_N_4_ for photocatalytic elimination of pollutants [19,24,41], photoreduction of carbon dioxide [42,43,44], water splitting for hydrogen production [28,45], and so on.

Recently, many researchers have reported the impact of different types of precursors on the structural, optical, morphological, and physicochemical properties as well as the efficiency of g-C_3_N_4_ photocatalysts. Enumerating some of the recent outcomes, Zhang et al. fabricated g-C_3_N_4_ by pyrolysis using an aggregate of urea and thiourea as precursors to enhance the specific surface area and separation of photo-induced electron-hole pairs, consequently improving photocatalytic activity of g-C_3_N_4_ photocatalysts [27]. Similarly, Kadam et al. successfully synthesized porous g-C_3_N_4_ nanosheets by heating low-cost urea and thiourea precursors at 520 °C for 2 h, and g-C_3_N_4_ showed the best photocatalytic efficiency for the elimination of cationic MB dye when exposed to visible light [23]. In addition, An et al. synthesized g-C_3_N_4_ materials by simply calcining mixtures of urea and thiourea precursors at 550 °C for photocatalytic degradation of RhB upon irradiation of visible light [46]. Guo et al. prepared g-C_3_N_4_ by polycondensation of melamine and urea at 550 °C for 3 h. For a melamine-to-urea molar ratio of 1:1, the photocatalytic degradation rate for MB was about 75% after 4 h of light exposure [47].

Similarly, calcination temperature control is one of the most common reaction parameters affecting the performance of the g-C_3_N_4_ photocatalyst. As a result, the synthesis of g-C_3_N_4_ by varying calcination temperature allows improvements in the surface area, crystallinity, extended absorption range of visible light, and overall enhanced photocatalytic activity [19,20,32]. Therefore, some recent studies have been carried out on the influence of calcination temperature on g-C_3_N_4_ photocatalytic performance. Enumerating a few of the outcomes, Fang et al. prepared porous g-C_3_N_4_ through direct pyrolysis of low-cost urea by controlling the condensation temperature in the range between 400 and 650 °C. In addition, the synthesized U-g-C_3_N_4_ had better photoactivity and long-term photocatalytic stability [20]. Similarly, Paul et al. prepared g-C_3_N_4_ by calcining urea at temperatures between 350 and 750 °C. It was discovered that increasing the calcination temperature to 550 °C enhanced the photocatalytic activity of g-C_3_N_4_ for the degradation of MB dye, whereas increasing the temperature beyond the optimum value decreased the photoactivity [32]. Moreover, Ke et al. synthesized g-C_3_N_4_ for the decomposition of rhodamine B (RhB) and methyl orange (MO) via thermal polycondensation of melamine at different calcination temperatures ranging from 450–700 °C. The g-C_3_N_4_ prepared at a calcination temperature of 700 °C revealed robust photocatalytic activity and stability against RhB and MO decomposition [19]. In addition, according to a review of the published literature in the field of visible-light-driven photocatalysis, the calcination temperatures for the fabrication of g-C_3_N_4_ using urea and thiourea were generally considered to be in the range of 520–550 °C and 450–650 °C, respectively [48].

Synthesis of the g-C_3_N_4_ photocatalyst using mixtures of stochiometric amounts of urea and thiourea at different calcination temperatures, especially in wastewater treatment using visible LED light irradiation, has not been reported in the previous literatures. Herein, in this work, we designed a novel single-step synthesis route to fabricate g-C_3_N_4_ nanomaterials by controlling various calcination temperatures of a mixture of stochiometric amounts of low-cost, environment-friendly urea and thiourea for complete photocatalytic degradation of RhB dye. Furthermore, the influence of varying calcination temperatures on the structural, optical, morphological, and physicochemical properties of g-C_3_N_4_ photocatalysts was carefully examined. Subsequently, the properties of the prepared g-C_3_N_4_ samples were characterized using several analytical techniques, and the photocatalytic activities of g-C_3_N_4_ samples were assessed by measuring the removal of rhodamine B (RhB) after irradiation of visible LED light.

## 2. Materials and Methods

### 2.1. Materials

Urea (CH_4_N_2_O, 99.5%), and thiourea (CH_4_N_2_S, 99%) were bought from Merck, Mumbai, India. Rhodamine B (99.9%) was obtained from Loba Chemie Pvt. Ltd., Mumbai, India. All reagents were used without further purification as received.

### 2.2. Methods

#### 2.2.1. Synthesis of g-C_3_N_4_ Samples

The preparation of g-C_3_N_4_ nanosheets was achieved by direct one-step calcination of mixtures of urea and thiourea in a muffle furnace under an air medium. Typically, weighed stoichiometric amounts of urea and thiourea were transferred to a 50 mL covered ceramic crucible with lid, and tightly wrapped in aluminum foil paper, then subjected to various calcination temperatures ranging from 500 to 650 °C in a muffle furnace for 3 h. Moreover, when the temperature is raised to 700 °C, nothing remains in the crucible, showing that the mixture of urea and thiourea has completely decomposed at calcination temperatures above 650 °C. Thus, the resultant powders were yellow. The resulting products were finely ground in a mortar and pestle after being cooled to room temperature. Finally, a series of light-yellow powders labeled g-C_3_N_4_-500 °C, g-C_3_N_4_-550 °C, g-C_3_N_4_-600 °C, and g-C_3_N_4_-650 °C were obtained and characterized using various analytical tools.

#### 2.2.2. Characterization Techniques

X-ray diffraction patterns of g-C_3_N_4_ samples were studied using X-ray powder diffraction (PanAlyticals, Almelo, The Netherlands) with Cu-Kα radiation (λ = 1.5406 Å), operating at 40 kV, and 15 mA. Diffraction intensity was continuously recorded over a 2θ range from 10° to 70° with a sweep step width of 0.01° and a scan rate of 10°/min. Fourier transform infrared (FTIR) spectroscopy (IR Tracer-100, Shimadzu, Kyoto, Japan) was used to probe functional groups in the range 400–4000 cm^−1^. The optical properties of the materials were evaluated with a UV-Vis diffuse reflectance spectrometer (UV-2600, Shimadzu, Kyoto, Japan) in the range 200–800 nm using barium sulfate (BaSO_4_) as background. Photoluminescence (PL) spectral data were obtained using a PL spectrophotometer (RF-6000, Shimadzu, Kyoto, Japan) at an excitation wavelength of 360 nm. The width of the excitation and emission slits was 10 nm, and the thermal stability of each g-C_3_N_4_ sample was obtained by TGA (Q600, TA Instruments, Hüllhorst, Germany). The morphology of the g-C_3_N_4_-600 °C sample was investigated by FE-SEM and HR-TEM.

#### 2.2.3. Photocatalytic Degradation Experiments

The photocatalytic activity of g-C_3_N_4_ photocatalysts with different calcination temperatures was evaluated by decomposing RhB as a model pollutant under visible-LED-light irradiation. In a typical experiment, 50 mg of g-C_3_N_4_ photocatalyst was dispersed in 100 mL of RhB solution (10 mg/L of RhB in 100 mL of distilled water). The suspension was stirred vigorously for 60 min in the dark before irradiation to ensure proper dispersion and establish adsorption–desorption equilibrium between the photocatalysts and RhB dye. The initial concentration (C_0_) of suspensions was obtained at adsorption–desorption equilibrium. A visible light source of (50 W, 5000 lm, 6500 k daylight, 220–240 V) LED lamp (Phillips, Kolkata, India) was used to initiate the photocatalytic reaction. At predetermined intervals, 3 mL of the solution was taken and centrifuged to separate the photocatalyst and to obtain the supernatant. Subsequently, the concentration of RhB in the supernatant was measured with a UV-Vis NIR spectrophotometer (Cary5000, Agilent, Santa Clara, California, USA) at its maximum absorption wavelength of $\left(\lambda \max=554\mathrm{nm}\right).$

The intensity change of the major absorption peak was used to assess dye degradation. The following equation was used to calculate photocatalytic degradation efficiency [32]:(1) Degradation Efficiency=1−CC0×100%=1−AA0×100%
where C_0_ represents the concentration of RhB dye at adsorption–desorption equilibrium (mg L^−1^), and C is the concentration of RhB at reaction time t; similarly, A and A_0_ represent corresponding values.

To solve the apparent degradation rate constant (k) of the RhB dye, a simplified Langmuir–Hinshelwood pseudo-first-order kinetic model was used [49]:(2) lnC0C=kt 
where C_0_ is the concentration of RhB dye at adsorption–desorption equilibrium (mgL^−1^), C is the concentration of RhB at reaction time t, k is the apparent reaction rate constant (min^−1^), and t is the irradiation time (min). Furthermore, the photolysis experiment was carried out following the same procedure without the addition of a catalyst.

## 3. Results and Discussion

### 3.1. XRD Analysis

X-ray diffraction patterns were used to identify the crystal structure of g-C_3_N_4_ prepared at various temperatures, as shown in Figure 1. The XRD peaks of g-C_3_N_4_ samples generated at calcination temperatures of 500, 550, 600, and 650 °C revealed two main diffraction peaks at around 13.1° and 27.2°. The characteristic peak at 27.2° represents interplanar graphitic stacking, while the peak at 13.1° represents inplanar structural packing, demonstrating g-C_3_N_4_ formation in all prepared g-C_3_N_4_ samples [42,50]. As the calcination temperature increases, the peak of g-C_3_N_4_ samples becomes sharper, while its width becomes narrower. As a result, an increase in peak sharpness indicated a higher degree of crystallinity of a photocatalyst [51]. Therefore, a higher degree of crystallinity of the photocatalyst accelerated the transfer of charge carriers to the surface, thereby inhibiting photoinduced electron-hole recombination by promoting the separation of charge carriers [19].

No solid sample was produced when the calcination temperature reached 700 °C during calcination, indicating that g-C_3_N_4_ completely decomposed into gases including CO_2_, H_2_S, NH_3_, and water. Therefore, at calcination temperatures above 650 °C, g-C_3_N_4_-prepared samples become thermally unstable, and optimal calcination could be between 600–650 °C [32]. In addition, Figure 1 revealed that the 2θ position for typical (002) peak of g-C_3_N_4_ samples prepared at calcination temperatures of 500, 550, 600, and 650 °C is 27.25, 27.08, 27.30, 27.19, and 27.25°, respectively.

Moreover, the crystallite size of synthesized g-C_3_N_4_ samples was determined using the Debye–Scherrer formula [52]:(3)$$\;D=\frac{k\lambda }{\beta \;\cos\theta }$$
where D is the crystallite size, λ is the wavelength of X-ray radiation (λ = 1.5406 Å), k is the Scherrer’s constant (0.94), θ is the diffraction angle, and β is the full-width half-maximum of the diffraction peak relating to the (002) plane. In this study, k was rounded to 0.9. Similarly, the interplanar spacing (d_hkl_) can be calculated as follows:(4)$${\;d}_{\mathrm{hkl}}=\frac{\lambda }{2\sin\theta }$$

Table 1 shows the crystallite size and interplanar spacing of all synthesized g-C_3_N_4_ samples at various temperatures corresponding to the (002) plane.

Table 1 demonstrates that as the calcination temperature rises, so does the intensity of the diffraction peak and the crystal size of the prepared g-C_3_N_4_ samples. A sharp intense peak at high calcination temperatures implies that the s-triazine layer arrangement is more regular and that there is a significant degree of aggregation [53].

### 3.2. Optical Properties Studies

Photoluminescence (PL), UV-Vis diffuse reflectance spectra (DRS), and FTIR spectra were used to investigate the optical properties of the prepared g-C_3_N_4_ samples.

#### 3.2.1. PL Analysis

Photoluminescence spectra are useful for understanding the separation and recombination of electron-hole pairs, which is an important mechanism in photocatalysis [54]. Lowering the PL intensity indicates a decrease in the rate of recombination of electron-hole pairs and that increases photocatalytic activity [19,55]. The PL spectra of g-C_3_N_4_ samples prepared at various calcination temperatures were obtained at 360 nm excitation wavelength. As shown in Figure 2, all prepared g-C_3_N_4_ samples exhibited an intense and broad photoluminescence peak between 437–440 nm and showed a slight red-shift at a temperature of 550 °C. A slight shift of PL peak was attributable to differences in the bandgap energy of each sample [47]. Except for the sample prepared at 550 °C, the PL intensity of all prepared g-C_3_N_4_ samples increased as the calcination temperature increased. This indicated that g-C_3_N_4_ prepared at 550 °C could have the lowest recombination rate of photoinduced electron-hole pairs [56]. In addition, an increase in the amount of tri-s-triazine content at a calcination temperature of 550 °C may lead to higher π-states and cause orbital overlap and reduce the PL intensity [48].

#### 3.2.2. UV-Vis DRS Analysis

Diffuse reflectance spectroscopy was used to study the optical properties of all prepared photocatalysts at different calcination temperatures, as shown in Figure 3a. The variation in the absorption edges of g-C_3_N_4_ samples was observed with changing calcination temperatures, as shown in Figure 3a. In addition, the variation of energy bandgap influences photocatalytic activity by affecting visible light absorption and the separation of e^−^/h^+^ pairs [32]. The bandgap energy (Eg) of the samples was calculated by using a Kubelka–Munk transformation, which can be expressed by the equation [8]:(5)$$\;{\left(F\left(R\right)\mathrm{hv}\right)}^{n}=A\left(\;h\nu -\mathrm{Eg}\;\right)$$
where $F\left(R\right)\;$is Kubelka–Munk function, h is the Planck constant, ν is the light frequency, A is a constant, and n is equal to ½ for direct bandgap and 2 for indirect bandgap materials. Generally, g-C_3_N_4_ is a well-known indirect bandgap semiconductor material [37]. Therefore, the energy bandgaps of g-C_3_N_4_ samples were calculated by extrapolating the linear portion of the $\left(F\left(R\right)h\nu \right)$^1/2^ versus $\left(h\nu \right)$ curve (as shown in Figure 3b). The energy bandgap of g-C_3_N_4_ samples prepared at different calcination temperatures of 500, 550, 600, and 650 °C was estimated to be 2.78, 2.75, 2.82, and 2.89 eV, respectively [37,47]. In comparison with calcination temperatures of 500, 600, and 650 °C, the bandgap at 550 °C was narrower. When the temperature further increased to 650 °C, this caused a blue shift in the absorption edge and an increase in band gap energy to 2.89 eV. The blue shift may be a result of a significant quantum confinement effect and the possible reason for better photoactivity of a sample obtained at 650 °C than at 500 and 550 °C [32].

#### 3.2.3. FTIR Analysis

FTIR spectroscopy is used to analyze the chemical compositions and functional groups of g-C_3_N_4_ samples synthesized at various calcination temperatures. As shown in Figure 4a, there were no significant differences in FTIR peaks at different calcination temperatures. All synthesized samples displayed prominent peaks in the wavenumber range of 1230–1740 cm^−1^, which corresponds to the stretching mode of conjugated heterocycles [1]. The stretching mode of C=N heterocycles is responsible for peaks of g-C_3_N_4_ samples seen at wavenumber about 1458, 1560, and 1630 cm^−1^. However, the C=O stretching mode at wavenumber 1740 cm^−1^ was observed in the g-C_3_N_4_ samples prepared at calcination temperatures of 500, 600, and 650 °C. FTIR peaks at wavenumber around 1230, 1319, and 1400 cm^−1^ belong to the aromatic C-N stretching mode, and also the peaks at wavenumber of 805 and 885 cm^−1^ resulted from the distinctive breathing mode of tri-s-triazine ring units. Moreover, the absorbed H_2_O molecules cause the occurrence of N-H and O-H stretching modes at broad bands situated in the wavenumber range of 3074–3320 cm^−1^ [57,58,59]. In Figure 4a, all peaks seen are the typical pattern of g-C_3_N_4_.

### 3.3. TGA Analysis

Thermal analysis of g-C_3_N_4_ samples was performed by heating them from ambient temperature to 700 °C in an air atmosphere. Figure 4b shows the thermal stability of g-C_3_N_4_ photocatalysts synthesized at various calcination temperatures. The weight loss is less at temperatures below 450 °C but increases as calcination temperatures decrease from 650 to 500 °C, which may be caused by the evaporation of water molecules that have been adsorbed on the surface of catalysts [60]. Meanwhile, the g-C_3_N_4_-650 °C sample did not lose weight significantly at a temperature below 450 °C, which is an indication of its great thermal stability. In addition, the significant weight loss of all prepared samples between temperatures 450 °C and 700 °C is attributed to the decomposition and condensation of g-C_3_N_4_ photocatalysts [42,61]. A thermogravimetric study shows that as-prepared g-C_3_N_4_ is non-volatile up to 600 °C and will be practically entirely degraded when the temperature climbs to 700 °C [36].

### 3.4. FE-SEM and HR-TEM Analysis

Figure 5a,b demonstrated low-magnification SEM and high-magnification TEM images of g-C_3_N_4_ synthesized at a calcination temperature of 600 °C. Figure 5a revealed that g-C_3_N_4_ prepared at 600 °C consists of large nanoflakes [46]. In addition, the formation of these nanoflakes could result in increased surface area which could lead to a large number of active sites, and a higher adsorption of pollutants over its surface [62]. Figure 5b showed a high magnification TEM image of g-C_3_N_4_ prepared at 600 °C, and it was almost transparent, which was due to its thin structure. The results indicated that g-C_3_N_4_ prepared at 600 °C was fluffy [56]. Therefore, we conclude that the enhanced photocatalytic activity is caused by the formation of nanoflake structures in g-C_3_N_4_ photocatalysts calcined at 600 °C.

### 3.5. Photocatalytic Activity Evaluation

The photocatalytic activity of rhodamine B for all the synthesized g-C_3_N_4_ samples was shown by plots of C/C_0_ vs. irradiation time (min), as shown in Figure 6. Without g-C_3_N_4_, the degradation rate of RhB was found to be almost insignificant, showing that the photolysis of RhB itself was negligible. Therefore, in the absence of a g-C_3_N_4_ photocatalyst, the rhodamine B dye solution was stable under visible LED light irradiation. The degradation of RhB was greatly improved when photocatalysts were introduced. As a result, the degradation of rhodamine B dye was caused by the presence of g-C_3_N_4_ in the solution.

The adsorption efficiencies of g-C_3_N_4_ photocatalysts prepared at 500, 550, 600, and 650 °C were about 1.85%, 0.66%, 3.5%, and 1.55% for RhB dye solution, respectively. The g-C_3_N_4_ sample prepared at a calcination temperature of 600 °C displayed the highest adsorption ability. Meanwhile, the photodegradation efficiency of g-C_3_N_4_-500 °C, g-C_3_N_4_-550 °C, g-C_3_N_4_-600 °C, and g-C_3_N_4_-650 °C was 18.34%, 22.2%, 94.83%, and 94.80%, after 180 min irradiation, as shown in Figure 6. The degradation efficiency and reaction rate constant of all prepared samples of g-C_3_N_4_ at various temperatures towards RhB dye removal are shown in Figure 6a,b, respectively, and are represented in Table 2.

Generally, Table 2 showed that boosted photodegradation efficiency of g-C_3_N_4_ photocatalysts was obtained as calcination temperatures vary between 600 and 650 °C. Furthermore, the crystallinity of the as-prepared samples increased at higher calcination temperatures.

Table 3 summarizes the findings of a comparison of the photocatalytic activities of g-C_3_N_4_-600 °C with other previously reported photocatalysts. As shown, g-C_3_N_4_-600 °C was a very potential photocatalyst and was extremely attractive for the actual treatment of industrial wastewater containing organic pollutants.

## 4. Conclusions

A visible LED light-driven graphitic carbon nitride photocatalyst was fabricated by single-step calcination of mixtures of urea and thiourea at different temperatures. Meanwhile, various analytical tools were used to study the structural, morphological, optical, and physicochemical properties of all prepared g-C_3_N_4_ samples. The presence of a typical g-C_3_N_4_ pattern was demonstrated by the XRD and FTIR spectra of all prepared g-C_3_N_4_ samples at various calcination temperatures. The influence of calcining temperatures on photocatalytic activities of g-C_3_N_4_ samples was investigated by photodegradation of RhB when exposed to visible LED light. The best photocatalytic activity of g-C_3_N_4_ photocatalyst was achieved at a temperature of 600 °C, due to its efficient separation of photoinduced charge carriers, high crystallinity, and enhanced visible light absorption range. Therefore, single-step synthesis of g-C_3_N_4_ using a mixture of urea and thiourea at high calcination temperature leads to the highest photocatalytic activity for the degradation of RhB dyes. This work can give a good insight into the synthesis of a cost-effective and efficient g-C_3_N_4_ photocatalyst at high calcination temperature for the treatment of industrial wastewater containing organic pollutants.

## Figures and Tables

**Figure 1 nanomaterials-13-00762-f001:**
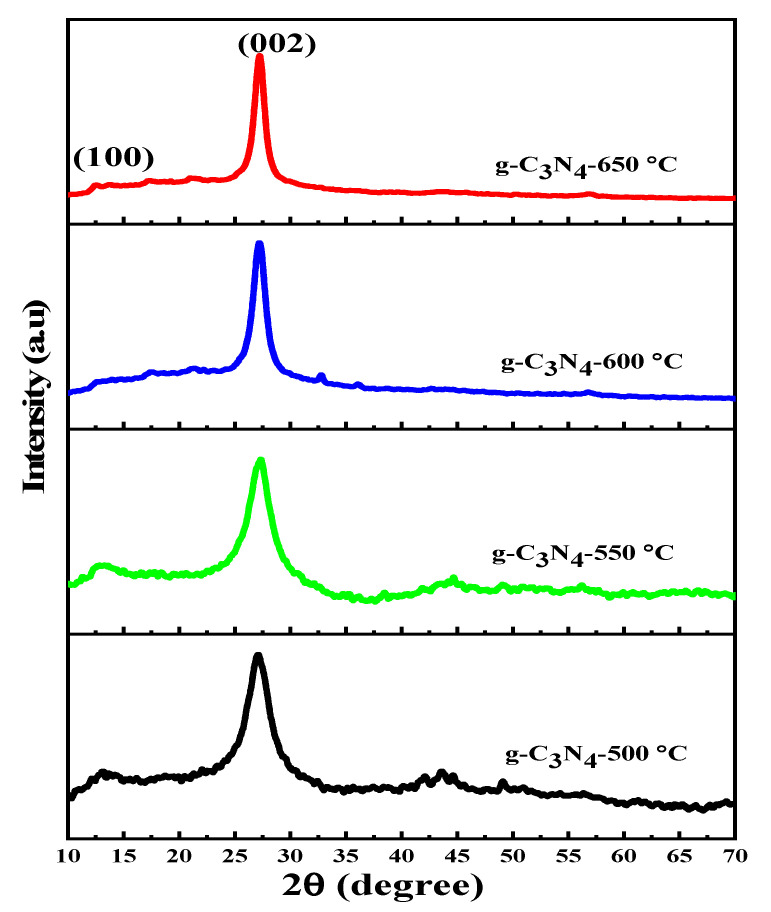
Calcination temperature-dependent XRD patterns of g-C_3_N_4_.

**Figure 2 nanomaterials-13-00762-f002:**
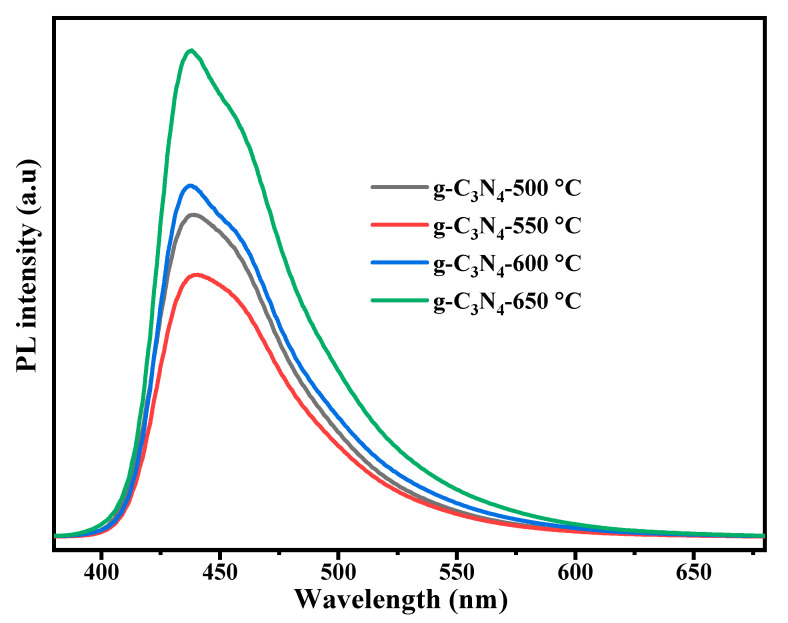
Photoluminescence spectra of g-C_3_N_4_ samples prepared at different calcination temperatures.

**Figure 3 nanomaterials-13-00762-f003:**
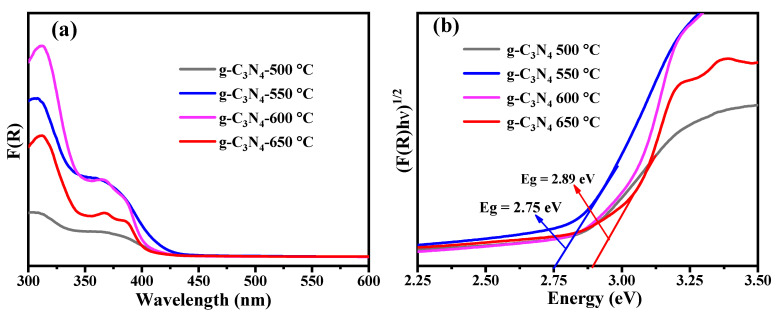
(**a**) UV-Vis DRS spectra and (**b**) corresponding Kubelka–Munk function plot of all prepared g-C_3_N_4_ samples.

**Figure 4 nanomaterials-13-00762-f004:**
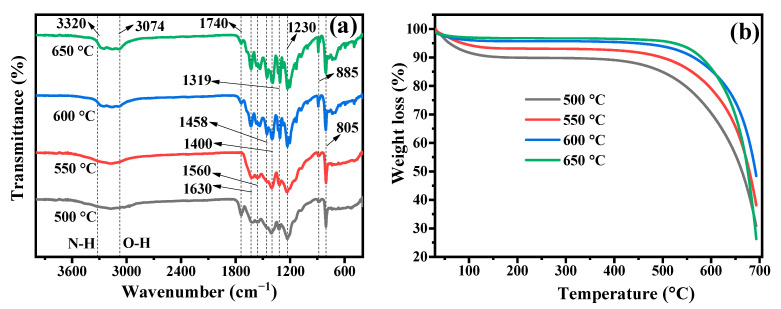
(**a**) FTIR spectra and (**b**) TGA curves of g-C_3_N_4_ photocatalysts prepared at different calcination temperatures.

**Figure 5 nanomaterials-13-00762-f005:**
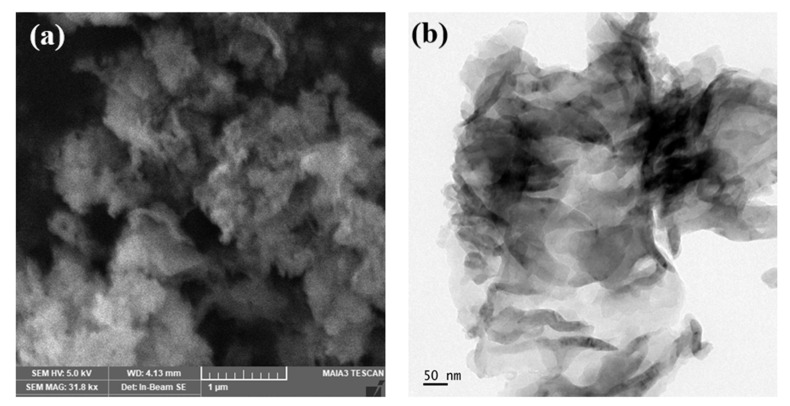
FE-SEM (**a**,**b**) HR-TEM images of g-C_3_N_4_ samples obtained at 600 °C.

**Figure 6 nanomaterials-13-00762-f006:**
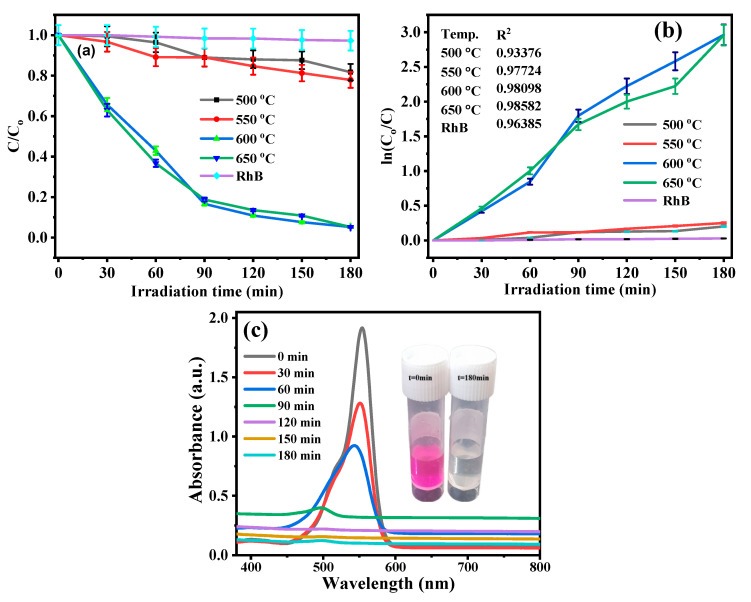
(**a**) Photoactivity of RhB of all prepared g-C_3_N_4_ photocatalysts under visible-LED light irradiation (**b**) kinetics of RhB degradation as function of ln(C_0_/C) vs. irradiation time of all prepared g-C_3_N_4_ photocatalysts and (**c**) UV-Vis spectra of RhB at different irradiation times over g-C_3_N_4_-600 °C (with inset image at t = 0 and t = 180 min).

**Table 1 nanomaterials-13-00762-t001:** Summary of structural properties and bandgap energy (eV) of all prepared g-C_3_N_4_ photocatalysts.

Samples	2θ (°)	β (FWHM) (Rad)	Crystalline Size (nm)	d_hkl_-Spacing (nm)	Eg (eV)
g-C_3_N_4_-500 °C	27.08	0.064	2.23	0.329	2.78
g-C_3_N_4_-550 °C	27.30	0.056	2.55	0.320	2.75
g-C_3_N_4_-600 °C	27.19	0.033	4.32	0.328	2.82
g-C_3_N_4_-650 °C	27.25	0.026	5.49	0.327	2.89

**Table 2 nanomaterials-13-00762-t002:** Photodegradation efficiency and reaction rate constant of all prepared g-C_3_N_4_ samples at different calcination temperatures.

S. No.	Photocatalyst	Degradation Efficiency	Adsorption Efficiency	Reaction Rate Constant (k) (min^−1^)
1	g-C_3_N_4_-500 °C	18.34%	1.85%	0.00113
2	g-C_3_N_4_-550 °C	22.20%	0.66%	0.00139
3	g-C_3_N_4_-600 °C	94.83%	3.5%	0.01646
4	g-C_3_N_4_-650 °C	94.80%	1.55%	0.01643

**Table 3 nanomaterials-13-00762-t003:** Comparison of photocatalytic activities of some selected photocatalysts with the present work.

Photocatalyst	Catalyst Dosage	Light Source	Irradiation Time	Pollutant Load	Degradation Efficiency	Ref.
U-g-C_3_N_4_	10 mg	300 W xenon lamp	25 min	RhB (0.25 μmol/L)	100%	[20]
CN-700 °C	100 mg	1000 W xenon lamp	30 min	RhB (100 mg/L)	99.11%	[19]
p–SCN–2	50 mg	1000 W Xe lamp	90 min	RhB (10 ppm)	97.5%	[63]
75:25SCN	0.12 g	100 W tungsten lamp	420 min	RhB (30 mg/L)	84.25%	[46]
g-C_3_N_4_-600 °C	50 mg	50 W LED lamp	180 min	RhB (10 mg/L)	94.83%	This work

## Data Availability

The data that support the findings of this study are available upon reasonable request from the authors.

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
