# Peer review of "Single-Step Synthesis of Graphitic Carbon Nitride Nanomaterials by Directly Calcining the Mixture of Urea and Thiourea: Application for Rhodamine B (RhB) Dye Degradation"

_nanomaterials, 2023, doi:10.3390/nano13040762_

Round 1

Reviewer 1 Report

The manuscript presents the synthesis and characterization of graphitic carbon nitride nanomaterials. The synthesis was performed by calcining a mixture of urea and thiourea at different temperatures. The properties of the prepared materials were assessed using several techniques and the removal of rhodamine B in an aqueous solution was used to determine the photocatalytic activity under visible irradiation.

The paper is well presented, and the methods are adequate and described appropriately, nevertheless, the results' discussion is somehow superficial and needs to be significantly revised for some characterizations.

The morphological properties are insufficiently described. Figure 5(a) is out of focus. The HR-TEM was not introduced in the materials and methods section. Only one of the prepared materials is investigated, but the presented data are not sufficient to draw any conclusions. The authors should either add significant information in the section or remove it completely (and any comments on the morphological properties) since it does not add any value to the presented work.

Also, the UV-Vis DRS analysis should be revised. I can't check the data but the red line in figure 3b does not follow the linear part of the spectrum as it should. Since the definition of the linear part of the spectrum is user dependent (especially when there are small additional absorption bands as in this case), the authors should add the used linear regressions at least as supplementary information.

In the PL analysis the reason why the sample calcined at 550°C behaves differently from the others should be addressed.

The sentence "and a high surface area" at line 348, should be removed if the authors won't perform a BET analysis.

Minor comments:

- Lines 63, 82, 116: references are unformatted;

- Lines 99, 143, 147: two words have a capital letter at the sentence beginning;

- All figures and tables references are represented by an error (MS word flair);

- When values are presented for all the samples, it is better to reference the values to the corresponding table and comment on the behavior instead of reporting all the values in the text (see lines 207, 263, 319, 322)

- Line 319, I think that MB should be RhB

Lines 354 to 357: in the authors contributions the experiments are not listed.

Author Response

The authors would like to thank the Reviewer for careful review of our manuscript.

Reviewer 2 Report

The work is about graphitic crbon nitride photocatalysts for cleaning waste water by photocatalytic oxidization of organic pollutants. The work is meaningful and interesting, but there are still some flaws which should be corrected as noted below, despite nice graphs some flaws significantly affect the presentation:

1.       There are two different citation methods please unify them.

2.       Cities of the chemical suppliers are missing.

3.       The authors write, that a 50 W LED lamp was used. Was the real power comsumption 50W, was light output 50W or was the light emission equal to a 50W thoungston lamp? I think it is better to write real power consumption or flux rate.

4.       Section 3.1 error reference not found.

5.       How were the gases described in methods section measured?

6.       Many broken references in this document please correct

7.       Page 6 section 3.1: Please start sentences not small and write full sentences.

8.       Section 3.2.1 : Please start sentences not small

9.       Komma after a point is not common in English…

10.   The authors omitted the presence of other inorganic highly efficient photocatalysts which also can clean waste waters[1] and even disinfect surfaces[2]. The authors should mention these systems and compare their advantages/ disadvantages. Especially since the light harvesting and production and structures are quite different.

11.   Reference 45 should since long been updated and contain the issue, and page numbers and volume. Please update the reference XXXXX

References

[1]        Z. Li, J. Shen, J.-Q. Wang, D. Wang, Y. Huang, J. Zou, CrystEngComm 2012, 14, 1874.

[2]        L. Mu, S. Rutkowski, T. Si, M. Gai, J. Wang, S. I. Tverdokhlebov, J. Frueh, Colloids Surfaces A Physicochem. Eng. Asp. 2021, 610, 125898.

Author Response

The authors would like to thank the Reviewer for careful review of our manuscript

Reviewer 3 Report

Manuscript deals with the Single-step synthesis of graphitic carbon nitride nanomaterials by directly calcining the mixture of urea and thiourea: Applica- tion for Rhodamine B (RhB) dye degradation. But publication point of view some modification necessary.

1. There are few grammatical mistakes. Please check the manuscript for grammar and English.

2. What is novelty of the present work? Rewrite it at the end of introduction section.

3. Compare your work (photocatalytic result) with other researcher work in tabular form.

4. Add error bar in fig.6 to estimate the error.

5. Add stability study of photocatalytic material.

6. Recycling experiments should be provided, and XRD or XPS should characterize prepared material after photocatalytic reaction.

7. Add XPS study of prepared material in order to confirm its oxidation states.

8. To enrich literature in the introduction section add some literature.

 i. Journal of Materials Science: Materials in Electronics 32 (11), (2021), 15577-15585, ii. Recent Patents on Nanotechnology 17 (1), (2023), 5-7, iii. Nanomaterials 13 (2), (2023),338, iv. Journal of Alloys and Compounds 928, (2022), 167133

Author Response

(The authors gave the same response as above.)

Reviewer 4 Report

In this manuscript, aiming the environmental pollution from dyes in the water, authors synthesized graphitic carbon nitride (g-C3N4) photocatalysts using mixtures of low-cost, environment-friendly urea and thiourea as precursors by calcination. Comprehensive characterizations have been performed. The results indicated that the rhodamine B (RhB) dye can be efficiently degraded by the photocatalysts. In general, the manuscript is well organized. However, there are still some issues to be addressed. A moderate revision is required before its acceptance.

1.     In the abstract, one or two sentences are suggested to show the background or aim of this work.

2.     More background on the water pollution should be provided with some recent supporting articles: Synthesis and Application of Granular Activated Carbon from Biomass Waste Materials for Water Treatment: A Review; Biochar derived from non-customized matamba fruit shell as an adsorbent for wastewater treatment; MOFs meet wood: reusable magnetic hydrophilic composites toward efficient water treatment with super-high dye adsorption capacity at high dye concentration; etc.

3.     More details on the raw materials should be provided, such as the purity.

4.     One scheme is suggested to show the whole experimental procedure, which is helpful for better understanding of readers.

5.     When introducing the various hazardous pollutants, specific references should be included: Polymers 14 (24), 5417, 2022; RSC Advances 12 (47), 30522-30528, 2022; Chemical Communications 56, 3935-3938, 2020; New Journal of Chemistry 46, 490-497, 2022; e-Polymers 22 (1), 285-300, 2022; etc.

6.     Error bars should be added to some of the figures for better scientific expression.

7.     More discussion on the formation mechanism of g-C3N4 should be provided.

8.     The references should be carefully rechecked to make sure full information is provided, such as the volume and pages.

9.     More comparison to the photodegradation by other materials should be provided with supporting articles, such as Hydrothermal Synthesis of Ce-doped ZnO Heterojunction Supported on Carbon Nanofibers with High Visible Light Photocatalytic Activity

10.  There are still some typos and grammar issues. Authors are suggested to recheck the whole manuscript.

Author Response

(The authors gave the same response as above.)

Reviewer 5 Report

1. The authors should correct the Introduction. The text at the end of Introduction should contain precise description of the main aims of the work and detailed tasks that will be completed.

2. The references should be corrected according journal requirements.

3. Moderate improvement of English text needed.

Author Response

(The authors gave the same response as above.)

Round 2

Reviewer 3 Report

The revision made by the author is satisfactory. The present form of the manuscript should be accepted.